# Regulation of Protein Post-Translational Modifications on Metabolism of Actinomycetes

**DOI:** 10.3390/biom10081122

**Published:** 2020-07-29

**Authors:** Chen-Fan Sun, Yong-Quan Li, Xu-Ming Mao

**Affiliations:** 1Institute of Pharmaceutical Biotechnology, School of Medicine, Zhejiang University, Hangzhou 310058, China; sunchenfan@126.com (C.-F.S.); lyq@zju.edu.cn (Y.-Q.L.); 2Zhejiang Provincial Key Laboratory for Microbial Biochemistry and Metabolic Engineering, Hangzhou 310058, China

**Keywords:** post-translational modifications, regulation mechanism, bacterial signaling, Actinomycetes

## Abstract

Protein post-translational modification (PTM) is a reversible process, which can dynamically regulate the metabolic state of cells through regulation of protein structure, activity, localization or protein–protein interactions. Actinomycetes are present in the soil, air and water, and their life cycle is strongly determined by environmental conditions. The complexity of variable environments urges Actinomycetes to respond quickly to external stimuli. In recent years, advances in identification and quantification of PTMs have led researchers to deepen their understanding of the functions of PTMs in physiology and metabolism, including vegetative growth, sporulation, metabolite synthesis and infectivity. On the other hand, most donor groups for PTMs come from various metabolites, suggesting a complex association network between metabolic states, PTMs and signaling pathways. Here, we review the mechanisms and functions of PTMs identified in Actinomycetes, focusing on phosphorylation, acylation and protein degradation in an attempt to summarize the recent progress of research on PTMs and their important role in bacterial cellular processes.

## 1. Introduction

Actinomycetes are Gram-positive filamentous bacteria, which grow mainly in the form of mycelia and propagate as spores. Members of this order show rich diversity in morphology, physiology and secondary metabolism. Actinomycetes were once considered as an intermediate group between bacteria and molds due to their morphology, with actinoid colonies, but they are not grouped into molds. Actinomycetes are grouped into 14 suborders based on 16S rRNA gene trees using molecular and chemical composition criteria, including Actinomycineae, Actinopolysporineae, Catenulisporineae, Corynebacterineae, Frankineae, Glycomycineae, Jiangellineae, Kineosporineae, Micrococcineae, Micromonosporineae, Propionibacterineae, Pseudonocardineae, Streptomycineae and Streptosporangineae [1]. Particularly, Streptomycetes are well-known as an important source for bioactive natural products, since they contribute about two-thirds of drugs with clinical, agricultural and biotechnological values. Most natural products are secondary metabolites, whose production is initiated during the switch from primary to secondary metabolism [2]. Nowadays, genome sequencing projects reveal a huge capacity for Actinomycetes to produce multiple secondary metabolites, since they contain numerous gene clusters responsible for natural product biosynthesis. However, most gene clusters are transcriptionally silent under laboratory conditions [3]. Thus, it is necessary to fully understand metabolic regulation in Actinomycetes, which will be the basis for better exploitation of natural products.

When environmental stresses such as nutrient starvations occur, substrate mycelium will autolyze and release the cellular contents. This is caused by uncontrolled action of murein hydrolases, prompting substrate mycelium into aerial mycelium formation, which is accompanied by the production of natural products [1,2,3,4]. In nature, the abundant secondary metabolites produced by Actinomycetes are considered a weapon to compete against other microorganisms for survival and secure their nutritional supply [5]. This response mechanism is basically divided into four processes, including extracellular signal stimulation, intracellular responses, signal transduction and target gene responses. By far, studies on metabolic regulation mechanisms are primarily focused on the transcription levels [6,7].

However, the functions of proteins can also be regulated via covalent modifications derived from some small chemical groups. These processes are called protein post-translational modifications (PTMs), which highly expand the chemical composition and information normally encoded in the amino acid sequence of the proteins [8,9]. In prokaryotic cells, although the number and degree of protein PTMs are much lower than eukaryotes and the level of modifications fluctuate greatly with environmental changes [8], there are increasing evidences that PTMs play a vital role in various cellular processes, such as protein stability [10], protein synthesis [11], spore formation [11], dormancy [12] and virulence [13]. The addition of modifications ranges from small chemical groups, such as acetyl groups (about 42 Da) or phosphate groups (about 80 Da), to more complex structures, including *N*-glycosylation (2–3 kDa) and polypeptide chains such as the prokaryotic ubiquitin-like protein (Pup) (about 8 kDa) (Table 1). While the functions of PTMs in primary metabolism are well-established, for example, their functions in cell division [11] and glucose assimilation [14], their roles in secondary metabolism remain elusive. With the development of two-dimensional gel electrophoresis and matrix-assisted laser desorption ionization time-of-flight (MALDI-TOF) mass spectrometry, the concept that PTMs regulate secondary metabolism was first proposed in 2002 [9]. The first analysis of a global modified proteome in Actinobacteria was reported for *Streptomyces coelicolor* (*S. coelicolor*) in 2011, showing that extensive protein phosphorylation occurs during cell differentiation [15].

PTMs remarkably increase the intricacies and flexibility of signaling networks [30]. It is known that the natural products in Actinomycetes are produced during specific cell cycle stages named secondary metabolism, after forming a vegetative mycelium that grows both on the medium and into it. This timely government emphasizes cell responsiveness to extracellular and intracellular stimuli [31]. The phosphoproteome is one of the most common bacterial PTMs, while acylation is a more complicated regulatory mechanism balanced by multiple roles of acyl-CoA for the biosynthesis of acyl-CoA-derived natural products [32]. In this general chemical biosynthetic pathway, acyl-CoAs serve as the donor for acylation, leading to enzymatic regulation, as well as the major precursors for metabolite production [33]. However, little is known about the relationship between PTM-mediated regulatory mechanisms and coordinated metabolic fluxes. In this review, we will summarize the major types of bacterial PTMs and provide an overview of their regulatory roles and molecular mechanisms involved in metabolism of Actinomycetes.

## 2. Phosphorylation

Phosphorylation, one of the best characterized PTMs, is the first type of PTMs found as a functional link between environmental nutrition and primary/secondary metabolism [15,33,34]. It involves attachment of a phosphate group to the amino acid side-chain of the protein active region, thereby changing the residue from hydrophobic to hydrophilic [35]. There are many donor sources of active phosphate groups and adenosine triphosphate (ATP) is the most used. Phosphorylation is an effective type of modification, changing the spatial structure of proteins. It is vital for coordinate gene functions for cell growth [11], virulence [36], biofilms [11] and quorum sensing [37]. After modification with a negatively charged phosphate group, proteins can be activated by exposing or stabilizing the active domain, thus incorporating the signal into protein sequence for subsequent transcriptional regulation.

Because there are different types of kinases, there are multiple pathways for phosphorylation to fulfill its regulatory functions in Actinomycetes (Figure 1A).

### 2.1. Two-Component Systems

Prokaryotic cells would adapt to environmental changes and respond by controlling their gene expression or by post-translational modifications, which would affect the activity or stability of proteins. In bacteria, transcription is mainly controlled by regulatory factors that interact with specific DNA elements during RNA synthesis, where the regulation of transcription initiation is most important during metabolic adaptation [6,38]. The main regulators capable of binding to DNA are called transcription factors (TFs), which inhibit or activate the transcription of target genes. Generally, TFs exert their action through a “single-component or two-component system (TCS)” to work, connecting specific types of environmental stimuli to transcriptional responses [39]. The two-component system is a signaling system that mainly utilizes phosphorylation of histidine and aspartate residues. Among 2758 bacteria whose genomes had been fully sequenced, 164,651 TCSs have been identified, with an average of 60 TCSs per genome [40]. In addition, the number of TCSs seems to be related to the number of environmental stimuli to which the organism is exposed [41]. Bacteria that live in stable environments have relatively few TCS genes, while others in frequently changing or diverse environments often contain a large number of TCSs. It has been estimated that *Streptomyces* contain a higher number of TCSs compared to other bacterial genera, which is considered to be the result of their soil-dwelling complex living environments. For example, 69 TCSs have been identified in the model strain *S. coelicolor*, which is up to 12.3% of the total ORFs in this strain (http://www.p2cs.org/).

TCSs generally consist of a histidine kinase (HK) with a domain that spans the plasma membrane and a response regulator (RR) located in the cytoplasm [41,42]. The transmitter domain of HK is conserved and contains two separate histidine kinase cytoplasmic domains, the N-terminal DHp (dimerized histidine phosphate transfer) domain and the C-terminal CA (catalyzed binding to ATP) domain. The response regulator protein contains one or more conserved N-terminal receiver domains and a variable C-terminal effector domain [43].

The mechanism of phosphorylation from TCSs can be divided into two steps. The first step is histidine phosphorylation. Once receiving an extracellular signal, histidine kinase self-phosphorylates the conserved histidine residue. The second step is aspartic acid phosphorylation. The phosphorylated sensing kinase activates the function of the Asp protein kinase, transferring the phosphate group from its His residue to the Asp residue of the response regulator. Thus, the signal is transferred from extracellular to intracellular compartments. In most cases, histidine kinases are bifunctional. In the absence of stimulated autophosphorylation, they can act as phosphatases for their homologous response regulators. Therefore, it is ultimately the ratio of kinase to phosphatase activity that regulates the output responses [44,45]. In some cases, the input signal may promote phosphatase activity rather than stimulate autophosphorylation [46].

Normally, phosphorylation activates the response regulator, most of which function through their DNA binding domain (DBD) [47]. In terms of these response elements, phosphorylation induces homodimerization, stimulates DNA binding and causes transcriptional changes [41]. Other common response regulators include diguanylate cyclases [48,49] and methyltransferases [50] in other bacteria. The main conserved TCSs are summarized in Table 2.

The PhoPR system is the most intensively studied TCS in *Streptomyces*, with a clear activation signal and a complete list of target genes [57]. PhoP is a master regulatory protein that functions in both primary and secondary metabolism. While phosphate limitation occurs in the environment, phosphorylated PhoR will phosphorylate PhoP and promote subsequent changes at the transcriptional level of a series of target genes. Results in *S. coelicolor* showed that the DNA binding capacity of PhoP DBD is stronger than that of the full-length protein [38]. It is possible that conformational structure of the target protein will change after phosphorylation on Asp residue. Phosphorylated Asp changes its charge from the positive to negative, leading to more exposure of the DBD region. In *S. coelicolor*, PhoP targets not only the genes that activate the phosphate metabolism system, but also two gene clusters for cell wall/extracellular polymer biosynthesis. In addition, PhoP represses gene expression for some unprecedented pathways under phosphate restriction, including nitrogen assimilation, oxidative phosphorylation, nucleotide biosynthesis and glycogen catabolism [57]. Moreover, PhoP has been shown to regulate many key genes in antibiotics production and morphologic differentiation, including *afsS*, *atrA*, *bldA*, *bldC*, *bldD*, *bldK*, *bldM*, *cdaR*, *cdgA* and *cdgB* [57]. It is noteworthy that in the PhoP-dependent *cpk* polyketide gene cluster, PhoP gathers at specific binding sites in the gene encoding the polyketide synthase [57]. This study showed that during phosphate starvation, PhoP plays an important role in inhibiting central metabolism, secondary metabolism and developmental pathways to cope with unfavorable environments.

### 2.2. Serine/Threonine/Tyrosine Phosphorylation

For a long time, protein phosphorylation in prokaryotes was thought to occur only on His and Asp residues of TCSs. Now, it has been demonstrated that prokaryotes show significant diversity in phosphorylation pathways. Numerous phosphorylation events occurring on serine/threonine residues are catalyzed by Hanks-type kinases (STKs) [64]. STK is a membrane or cytoplasmic protein with additional subdomains responsible for regulating STK activity or affecting its subcellular localization [65]. The structure of the catalytic domain in different kinases is significantly conserved in both Eukaryotes and bacteria. STK activation occurs through phosphorylation of at least one Ser/Thr residue in the activation loop [66]. Besides the catalytic domain, many bacterial STKs also contain other domains that mediate ligand binding and/or protein–protein interactions [67].

The large majority of protein-tyrosine kinases characterized in bacteria belong to the family of bacterial protein-tyrosine kinases (BY-kinases) [68], which have thus far only been encountered in bacteria [69]. Their active sites are located in the catalytic domain, which can bind ATP and transfer its gamma phosphate to the hydroxyl unit of the tyrosine residue [70]. Both of these two kinases act in the regulation of various signal transitions via substrate phosphorylation [66,68,69].

Attempts have been made to explain the mechanism of phosphorylation by analyzing the structure of the STK. The first bacterial STK for which the structure has been characterized by crystallization is protein kinases B (PknB) from *Mycobacterium tuberculosis* (*Mtb*) [71,72]. PknB is essential for growth of *Mtb* and has been proposed as a possible drug target [73]. The proteins regulated by PknB are involved in peptidoglycan synthesis and renewal, cell division, lipid metabolism, translation and central carbon metabolism. Loss of PknB leads to growth arrest, cell shortening and a significant increase in sensitivity to β-lactam antibiotics and rifampicin, the major anti-tuberculosis drug [73].

PknB mutant is a portion of PknB with kinase activity in the absence of regulatory cues. The crystal structure of PknB mutant shares a similar structure with activated states of eukaryotic Ser/Thr protein kinases [71]. The crystal structure revealed that the activation loop, which contains four of the phosphorylated residues among the total of six phosphorylated sites present in PknB, is disordered, similar to the characteristics of inactive kinases [71]. In the presence of ligand muropeptides, two or more PknB monomers bind to a ligand molecule through their extracellular domain (Figure 1A). This action brings the catalytic domains of adjacent molecules closer together, resulting in the formation of symmetrical back-to-back dimers, and thus activating the kinase through autophosphorylation [71]. In addition, heterodimers composed of inactive PknD mutant containing a normal dimer interface and wild-type monomers can still phosphorylate wild-type catalytic domains, consistent with the allosteric activation of kinases after dimerization [74]. It is unlikely that autophosphorylation after dimerization occurs through intermolecular phosphorylation. Meanwhile, ligand-promoted dimerization cannot be the only activation mechanism of bacterial STKs. Activated kinases can also directly phosphorylate downstream protein targets or activate soluble kinases by forming asymmetric front-to-front dimers, which then phosphorylate downstream targets as a part of the signaling pathway [75]. The co-crystallized structure of the kinase domain of PknB with the inhibitor molecule revealed a second dimerization interface, in which the two monomers bound to the inhibitor to form an asymmetric front-to-front dimer (Figure 1B) [75]. The conformation of these two proteins indicates that one monomer acts as an activator and the other monomer acts as a substrate. This dimerization provides a new mechanism by which allosterically activated kinases can be phosphorylated, thereby activating other kinases that are not related to the receptor domain.

In addition, phosphorylated proteins can also promote temporary protein–protein interactions to participate in signal transduction, as exemplified by KbpA. KbpA is an inhibitory protein of AfsK occluding the autophosphorylation site of the latter by combining with it [76]. AfsK-AfsR-AfsS is one of the first regulatory cascades involving Ser/Thr kinases that was confirmed in bacteria [76]. The AfsK-AfsR system in *S. coelicolor* A3(2) globally controls secondary metabolism [76]. AfsK is a kinase that loosely attaches to the membrane. The crucial step in its activation is autophosphorylation on Ser and Thr residues and subsequent phosphorylation on Ser and Thr residues of the pleiotropic regulator AfsR. Studies in *S. coelicolor* showed that phosphorylation of KbpA enhanced its binding effects with AfsK [77].

The universality and diversity of kinases in the bacterial community highlight their role in the adaptation of bacteria to various environmental stimuli. A series of available knockout tools, including the CRISPR-Cas9 system [78], facilitate the functional study of phosphorylation in metabolism. However, through reviewing previous studies, the real challenges appear to lie in two aspects: determining the activation signals and turning theoretical research into applications and techniques.

## 3. Acylation

It is clear that PTMs are an essential regulatory mechanism in eukaryotic cell signaling. Lysine acylation is emerging as a ubiquitous and conserved PTM in living cells. Recent developments in proteomics have revealed precise and large-scale examples of acylation in various bacteria. They are found in almost every metabolic pathway, including primary and secondary metabolism. It has been demonstrated that acetylation on lysine residues is a nonspecific enzymatic reaction or spontaneous non-enzymatic ligation, and the level of acetylation is ultimately dependent on the concentration of acetyl-CoA or acetylphosphate (AcP) (Figure 2A) and the activity of acetyltransferases and deacetylases.

### 3.1. How Acetylation and Deacetylation Occur

Among the wide range of PTMs, acetylation is a prominent regulatory mechanism, with acetyl-CoA functioning as the major active acyl-donor (Figure 2B). Acetylation was extensively studied and it is highly conserved among all species ranging from bacteria to humans. Previous work has reported that *Ma*Kat (from *Micromonospora aurantiaca*) is an acetyltransferase of acetyl-CoA synthetase, containing an amino acid-binding (ACT) domain at the N-terminus of the catalytic GNAT domain [79]. The ACT domain not only confers acetyl-transferase activity to bind to the peptide, but also has the characteristics of amino acid-induced allosteric regulation. It is clear that the proper spatial structure is of vital importance for the activity of protein acetyltransferase. For instance, the activity of protein acetyltransferases (PatA) closely depends on the intracellular amount of Asn and Cys, which finding broadens our understanding of protein acetylation in response to nutrient availability [80].

The ACT domain shows promiscuity to the motif of modified sites, while the removal of acetyl groups is regulated more elaborately. There are two types of deacetylases in Actinobacteria, Zn^2+^-dependent histone deacetyltransferases (HDACs) and the NAD^+^-dependent deacetylase CobB. In eukaryotic cells, it has been shown that different classes of HDACs display versatility in the use of substrates involved in lysine deacylation, where deacetylase activity is robust in all HDACs [81,82]. The Sir2-like enzyme CobB is the best-studied deacetylase in bacteria and possesses the most diverse deacylase activity by far [14,18,83,84,85]. Besides acetyl groups, studies have shown that CobB can also recognize succinyl [83], propionyl [32], de-2-hydroxyisobutyryl [85] and crotonyl [14] groups. This promiscuity in the selection of acyl groups ensures that the eraser is simple and efficient, adapting to the small genome capacity in prokaryotic cells.

Little is known about the substrates and the specificity of Zn^2+^-dependent lysine deacetylases, such as the AcuC deacetylase in *Bacillus subtilis*. It has been reported that in eukaryotic cells, metal-containing deacetylases (class I, II and IV HDACs) have different affinity to diverse acyl groups. In particular, class I HDACs are the major histone decrotonylase, which includes HDAC1, the only metal-containing deacetylases that prokaryotic cells possess [82].

Recent work in *E. coli* revealed that protein acetylation also occurs through non-enzymatic pathways, where the reactive high-energy metabolite AcP plays an important role in protein acetylation. Different from enzymatic catalysis, the non-enzymatic pathway is indiscriminate, occurs slowly, and results in low-levels of acetylation [86]. Intriguingly, the properties of nonenzymatic acetylation sites revealed little relevance to enzyme-catalyzed acetylation, which may be determined by surface accessibility, three-dimensional microenvironments and the pKa value of lysine [87]. Most putative GNATs in *E. coli* are known to catalyze acetylation on nonprotein substrates or N-terminal acetylation of proteins, thus decreasing the probability of promiscuous acetyltransferase activity [88]. However, it is possible that AcP may work as a cofactor for an undiscovered acetyltransferase to acetylate numerous sites. Importantly, CobB can deacetylate acetyllysines generated via both AcP and acetyltransferases [89].

### 3.2. The Roles of Acetylation in Actinobacteria

Analysis of mass spectrometry-based acetylproteomics after affinity enrichment with antibodies, shows the variation of protein sequences in various organisms under different physiological states, thereby revealing not only the modification sites, but also the degree of modifications. Most acylation occurs on lysine, which potentially changes the charge of modified residues and the spatial structure of the protein, thereby affecting its functional activity. In recent years, with the development of natural product research, researchers began to focus on the functional significance of acylation in the Actinobacteria, hoping to resolve bottlenecks of mining and yielding of secondary metabolites.

#### 3.2.1. Acetylation Functions in Cellular Signaling for Nutrient Assimilation

There is abundant evidence to link nutrient sensing with protein acetylation. Acetyl-CoA is the main active donor for acetylation and is also a pivotal substrate in the metabolism of energy substances. The fluctuation of external nutrients will inevitably change the concentration of cellular acetyl-CoA, which directly affects the degree of acetylation [83]. Changes in the degree of modification will result in different physiological outputs, depending on the function of the substrate protein, which will further in turn lead to the flux of modification levels. It has been demonstrated that crotonylated glucose kinase could enhance carbon catabolite repression to redistribute the utilization of carbon sources [14]. It is noteworthy that the donor of crotonylation, crotonyl-CoA, is mainly synthesized from acetyl-CoA [90]. Moreover, alterations in nutritional conditions can also activate the expression of many important global regulatory elements. It is something of a paradox that in some cases the regulator may regulate the transcription of enzymes in different functions simultaneously, causing opposing results. For instance, in response to nitrogen starvation, GlnR is overexpressed and directly binds to the promoters of genes encoding deacetylases, thereby acting as a transcriptional activator in *Saccharopolyspora erythraea* (*S. erythraea*) and *S. coelicolor* and as a repressor in *M. smegmatis* [91]. Furthermore, GlnR also controls the gene expression of acetyl-CoA synthases, including *acs1*, *acs2*, *acs3* in *S. erythraea*, thus increasing the source of acetyl donors (Figure 2C) [92].

A large number of acetylated proteins have been identified in association with pathways of carbon metabolism. Given that acetyl-CoA serves as an indispensable intermediate, it may play a bridge role in the acetylation and metabolic status, such as in the tricarboxylic acid cycle and metabolite biosynthesis, and acetylation may participate as a feedback mechanism to regulate carbon source fluctuations.

#### 3.2.2. Acetylation Plays a Key Role in Secondary Metabolism

It is common in prokaryotic cells for acetylation to be closely linked to primary metabolism and translation, as acetylation accumulates in the stationary and sporulation phases in liquid and solid cultures, respectively [10]. In recent years, due to interests in diverse natural products, researchers have focused on the relationship between acetylation and secondary metabolism. The earliest acetylome characterized in *Streptomyces* implied the possibility that acetylation functions in secondary metabolism regulation, and the first detailed study was carried out on *S. griseus* StrM (Figure 2D), a deoxysugar epimerase in the biosynthesis of streptomycin [10]. MS/MS data showed the major acetylated site is Lys70, which is conserved among deoxysugar epimerases. Modification on this site would abolish its catalytic capacity and reduce its stability in cells, resulting in restriction of streptomycin biosynthesis. However, the stimulus signal for acetylation on Lys70 is still unclear [10].

PTM of important biosynthetic enzymes of secondary metabolites would limit their production [10]. The concentration of acyl-CoA could directly dominate the levels of acylation, while these short-chain fatty acids attached to CoA are vital precursors for biosynthesis of various natural products (Figure 2E) [32]. It is necessary to determine which mechanisms interplay to balance the restriction and impetus caused by the acyl-CoAs. In Actinobacteria, mutants with higher yields of secondary metabolites can increase the efficiency of the use of acyl-CoA in biosynthesis [32]. In other words, their metabolic process can transport more acyl-CoA to the corresponding metabolites instead of to the acylated proteins. However, oversupply of acyl-CoAs in wild-type strains will result in hyperacylation and, subsequently, decrease in production of secondary metabolites. This is a kind of carbon overflow, and we also suppose it is a carbon source storage, since under appropriate conditions, acyl-CoA can be reused to reverse this reaction.

*Saccharopolyspora erythraea* is the producer of erythromycin, an antibiotic widely used for treatment of infections caused by a variety of Gram-positive bacteria. 6-deoxyerythronolide B (6-dEB) is the macrocyclic aglycone of erythromycin, which is composed of one propionyl-CoA and six methylmalonyl-CoA molecules. Propionyl-CoA, the starter unit, is an important regulator in the erythromycin biosynthetic pathway. In industry, the supply of propionyl-CoA usually increases by the addition of propionate or *n*-propanol, which is mainly catalyzed by acyl-CoA synthases [32].

PTMs function in regulation of the catalytic activity of acyl-CoA synthases. Studies showed that the high level of propionyl-CoA-induced propionylation on acetyl-CoA synthase (Acs) inhibits its activity to synthesize propionyl-CoA and causes an imbalance between the formation and assimilation of propionyl-CoA [18]. Researchers found four propionyl-CoA synthases in *S. erythraea*. Three of them were controlled by propionylation while SACE_1780 resisted propionylation. Through knocking out the acyltransferase-encoding gene *acuA* or overexpressing SACE_1780 to release feedback inhibition caused by propionylation, the yield of erythromycin was 10% or 33% higher, respectively [93]. These studies showed that PTMs play a key regulatory role in the regulation between precursor supply and product biosynthesis.

The link between acylation and metabolism is evident. Both enzymatic (acetyl-CoA) and non-enzymatic (acetylphosphate) modification mechanisms use primary metabolites as active acyl donors. Changes in the nutritional environment often trigger fluctuations in the level of intracellular acylation. Acylated metabolic enzymes in the central metabolism may serve as a feedback mechanism to guide nutrient assimilation. In the secondary metabolism, especially the biosynthesis of acyl-CoA-derived natural products, the balance between acyl-CoA in acylation and biosynthesis works to stabilize the concentration of metabolites.

## 4. Pupylation

Protein renewal is one of the foundations for all living organisms to maintain normal cell physiology. Proteasomes are ubiquitous in eukaryotic cells, and most of the proteins degraded by proteasomes are labeled with ubiquitin. Many bacteria are deficient in proteasomes, while they have typical proteases for protein turnover. In 2008, Pearce et al. found a protein similar to ubiquitin in *Mtb*, which was named prokaryotic ubiquitin-like protein (Pup) [94]. Pup targets a variety of functional proteins under the action of cofactors, which are degraded through the proteasome. Therefore, Pup acts as a degradation signal and directly guides proteins to the proteasome for degradation. Proteasomes of prokaryotes have been found only in Actinomycetes [95], where *M. tuberculosis* is one of the few bacteria known to have proteasomes on which relatively detailed research about mechanism and function has been performed. Its target proteins are involved in multiple aspects, such as substance metabolism in carbon metabolism and fatty acid metabolism, signaling pathways, toxic and antitoxic factors and cell wall and cell membrane components, and are related to the pathogenicity of *Mtb* [27]. Therefore, its proteasome is considered a new drug target for *Mtb* treatment.

Studies in *S. coelicolor* showed that pupylation-deficient mutant is more sensitive to the oxidants cumene hydroperoxide and hydrogen peroxide than the wild-type strain [96]. The researchers speculated that pupylation may be a type of response to the destruction of proteins caused by oxidants and/or other forces, affecting protein activity or localization rather than half-life [97]. Furthermore, *Δ pup* and *Δ pafA* (deficient of proteasome accessory factor A) mutants have defects in both spore formation and secondary metabolism, while *Δ prc* (deficient of proteasome components PrcA and PrcB) mutant does not, suggesting that pupylation does not only work by degrading proteins, but also acts by PTM in *Streptomyces* [97].

*Mtb* proteasome studies showed that pupylation is not only necessary for *Mtb* to survive and replicate in the host, but also importantly, it is essential for its survival in the host in a non-replication state. *Mtb* mainly inhabits host macrophages after infection. Studies have shown that pupylation and the proteasome are key elements that prevent the removal from host macrophages [27,98]. Furthermore, in the *Mtb* proteasome pathway, Pup serves as a protein modifier and provides an important means for identifying and characterizing *Mtb* substrates targeted to the proteasome. The characterization of such substrates will differentiate the *Mtb* pathways involved in their survival in vivo and form the basis for rational drug design [99]. In fact, chemical inhibitors (rhodanines) have been identified to inhibit dihydrolipoamide transferase, an enzyme for *Mtb* to resist to the effects of reactive nitrogen intermediates. It is foreseeable that the *Mtb* proteasome pathway is an exciting area for discovering new *Mtb* drug targets [100].

Among the Actinomycetes, research on the structure of the proteasome are mainly concentrated on *M. tuberculosis*. Similar to eukaryotic cells, prokaryotic Pup-proteasome-mediated protein degradation requires a series of cofactors to function, including deamidase of Pup (Dop), ligase proteasome accessory factor A (PafA) and mycobacterial proteasome ATPase (Mpa) (Figure 3). However, by comparing the sequence structure of the labeled proteins, it was found that there was no obvious homology between them [26]. Therefore, how PafA recognizes substrates destined for pupylation needs further investigation. In eukaryotic cells, other parallel modifications like methylation [101] on substrate proteins, play a key regulatory role in protein tagging for degradation and ubiquitination may also function as a signal for protein transport [102]. This provides a research direction for the mechanism in prokaryotes.

It is noteworthy that although ubiquitination does not exist in the intracellular system of *Mtb*, it has an important regulatory effect on the pathogenicity of *Mtb*. *Mtb* can use a variety of strategies to interfere with the normal biologic functions of the hosts to achieve immune escape. Meanwhile, the host cells also have a variety of anti-infective immune defense mechanisms for resisting and clearing invaders. For example, an important virulence factor Rv0222, a secretory protein of *Mtb*, can only recruit anti-inflammatory protein molecules after being ubiquitinated by the host, thereby blocking the host’s anti-tuberculosis immune pathway [98]. However, ubiquitin-mediated xenophagy is also an important inherent immune defense mechanism for the host to clear various intracellular pathogens, including *Mtb*. Another study found that the Rv1468c protein located on the surface of *Mtb* can bind to ubiquitin chains and further recruit autophagy-related proteins, which ultimately leads to the formation of autophagosomes and the removal of pathogenic bacteria [103].

## 5. Conclusions

It is clear that the link between PTMs and primary metabolism is intimate. Most of the donors for modification are derived from primary metabolites, like acetyl-CoA and ATP. In addition, sir2-dependent deacylation also requires the presence of the NAD^+^/NADH redox pair for catalysis, which plays a key role in central metabolism. Most of the enzymes in primary metabolism have at least one type of modification, which reversibly regulates their functional activities. In summary, PTMs are an important manner for negative feedback regulation in primary metabolism.

In the process of secondary metabolism, the functions of PTMs need to be further explored. On one hand, PTMs directly modify enzymes in the biosynthetic pathway responding to changes in concentration of the donor, reducing their catalytic activity or altering protein stability. On the other hand, acyl-CoAs and their derivatives are synthetic precursors of certain secondary metabolites and involved in their biosynthesis. Therefore, the metabolic flow of acyl-CoAs also directly affects the yield of desired products.

Herein, we summarize various PTM functions in Actinobacteria and how they work. The exciting research mentioned here revealed their general regulatory mechanisms in primary and secondary metabolism. It not only enables the bacteria to respond quickly to changes in the external environment, but also balances the strength of various metabolic pathways. At the same time, metabolic engineering based on the principle of PTMs also provides new ideas for future research in synthetic biology. However, the regulatory mechanism of PTMs in cells still has many unanswered questions, such as substrate recognition mechanism of acyltransferases and regulatory mechanism to balance between formation and utilization of cellular acyl-CoAs. In the coming years, we can expect more interesting discoveries about PTMs in Actinobacteria, specifically related to the development of secondary metabolites for industrial and therapeutic uses.

## Figures and Tables

**Figure 1 biomolecules-10-01122-f001:**
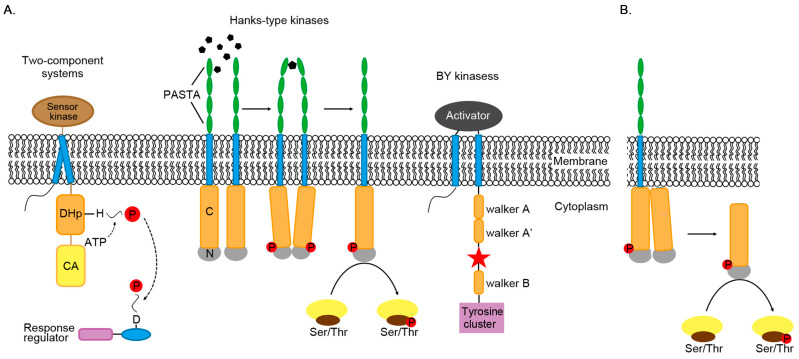
Phosphorylation mechanism of three types of kinases. (**A**) Three types of protein kinases catalyze the transfer of phosphate groups (P) to histidine (H), aspartic acid (D), serine, threonine and tyrosine. Most of these kinases are bound to the membrane. Extracellular domain PASTA (penicillin-binding protein and serine/threonine kinase-associated) binds peptidoglycans and induces the intracellular catalytic domains closer, resulting in activating kinase domains by autophosphorylation. After phosphorylation, two monomers dimerize through the back sides of the N-terminal lobes (N). BY kinase will interact with its activator to stabilize its ATP-binding domain. The red star is the phosphorylation site. DHp is the abbreviation of dimerized histidine phosphate transfer domain, and CA is the abbreviation of catalyzed binding to ATP. (**B**) In STK, activated kinases can directly phosphorylate target proteins.

**Figure 2 biomolecules-10-01122-f002:**
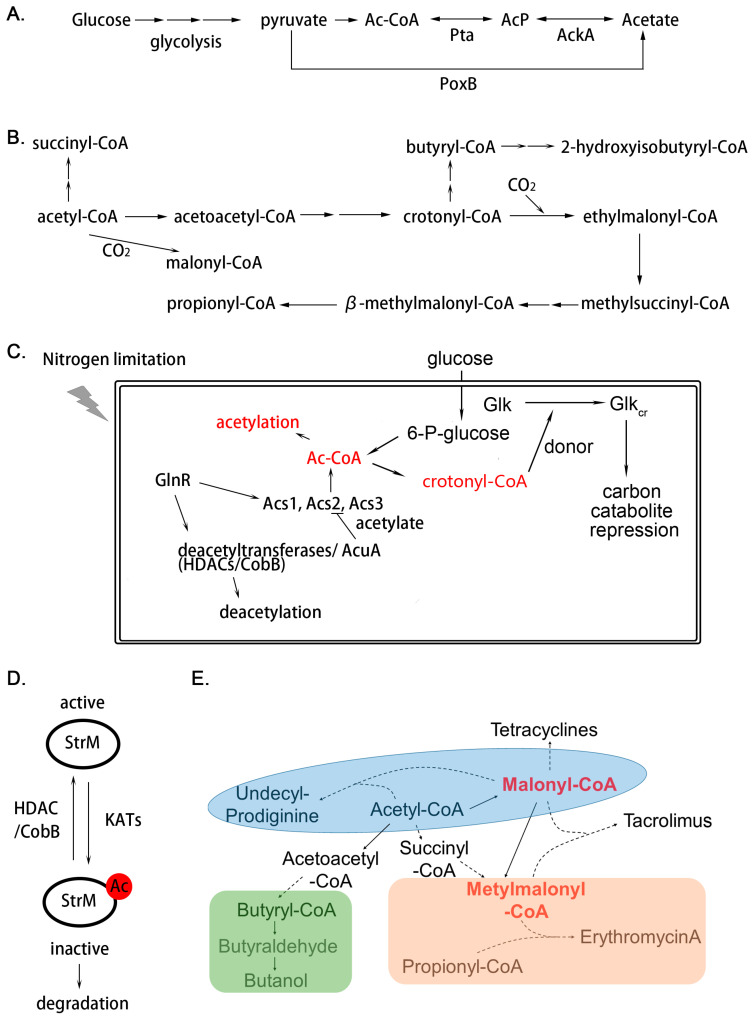
Regulation mechanism of acylation in Actinomycetes. (**A**) Source of intracellular acetyl-phosphate (AcP) synthesized from glycolysis. In bacteria, acetate can be interconverted to acetyl-CoA (Ac-CoA) through acetate kinase (AckA) and phosphotransacetylase (Pta) reversibly. The pyruvate oxidase (PoxB) is another way for acetate synthesis; (**B**) biosynthetic pathway of Acyl-CoAs derived from acetyl-CoA; (**C**) regulation mechanism in the terms of acylation responding to extracellular nutrient fluctuation; (**D**,**E**) two types of regulation mechanisms in secondary metabolism through acylation. Protein acylation would result in suppression of enzyme activity and decreased stability; (**D**) In secondary metabolites biosynthesis, acyl-CoAs also act as important precursors of diverse acyl-CoA-derived natural products; (**E**) solid line indicates that the product is synthesized directly; the dotted line indicates indirect synthesis.

**Figure 3 biomolecules-10-01122-f003:**
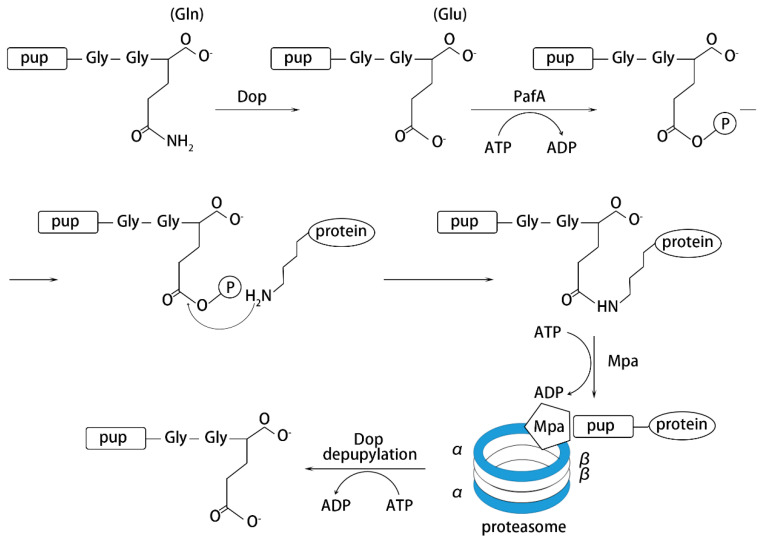
Process of pupylation-induced protein degradation.

**Table 1 biomolecules-10-01122-t001:** Main types of PTMs in Actinomycetes.

Organism	Main Antibiotic	PTM	Functions	Reference
*Streptomyces coelicolor*	actinorhodin and clorobiocin	Phosphorylation	Modulating differentiation and secondary metabolism	[11]
*Mycobacterium tuberculosis*	none		Virulence, fatty acid biosynthesis and two-component regulatory system	[16]
*Mycobacterium smegmatis*	none		Environmental adaptation, including dormancy and drug resistance	[17]
*Streptomyces roseosporus*	daptomycin	Acetylation	Governing cellular processes, including secondary metabolites biosynthesis	[18]
*Streptomyces coelicolor*	actinorhodin and clorobiocin		Governing cellular processes, including secondary metabolites biosynthesis	[9]
*Saccharopolyspora erythraea*	erythromycin		Central metabolism like protein synthesis, glycolysis, citric acid (TCA) cycle and a direct regulation in erythromycin synthesis	[19]
*Mycobacterium tuberculosis*	none		Metabolism, persistence and virulence	[20]
*Saccharopolyspora erythraea*	erythromycin	Malonylation	Central metabolism and erythromycin biosynthesis	[21]
*Streptomyces coelicolor*	actinorhodin and clorobiocin	Succinylation	Protein biosynthesis and carbon metabolism	[22]
*Mycobacterium tuberculosis*	none		Resistance to antibiotics	[23,24]
*Streptomyces roseosporus*	daptomycin	Crotonylation	Governing cellular processes, including carbon catabolite repression and secondary metabolites biosynthesis	[14]
*Mycobacterium tuberculosis*	none	Glutarylation	Governing protein folding and metabolic process related with stress reaction	[25]
*Streptomyces coelicolor*	actinorhodin and clorobiocin	Pupylation	Protein degradation	[26]
*Mycobacterium tuberculosis*	none		Substance metabolism, toxic and antitoxic factors, cell wall and cell membrane components and pathogenicity	[27]
*Streptomyces coelicolor*	actinorhodin and clorobiocin	O-glycosylation	Maintaining cell wall integrity and regulating enzyme function	[28]
*Streptomyces coelicolor*	actinorhodin and clorobiocin	ADP-ribosylation	Morphologic differentiation and antibiotic production	[29]

**Table 2 biomolecules-10-01122-t002:** The conserved two-component systems (TCSs) in Actinomycetes.

TCSs	Organism	Function	Reference
MacRS	*S. coelicolor*	Aerial mycelium formation/membrane integrity and/or other membrane-associated activities	[51]
MtrAB	*M. tuberculosis*	DNA replication and cell division	[52]
	*S. venezuelae*	Antibiotic production, nutrient assimilation and aerial mycelium formation	[53]
DraRK	*S. coelicolor*	Antibiotic production	[54]
TunRS	*S. coelicolor*	Cell wall metabolism and *tmrB-like* gene regulation	[55]
CssRS	*S. lividans*	Misfolded protein regulation	[56]
PhoPR	*Streptomyces*	Phosphate assimilation and secondary metabolism	[57]
AbrC1/2/3	*S. coelicolor*	Antibiotic production	[58]
EsrSR	*S. coelicolor*	Cell envelope stress response	[59]
AfsQ1/2	*S. coelicolor*	Antibiotic production	[60]
OsaABC	*S. coelicolor*	Osmotic stress response	[61]
GluRK	*S. coelicolor*	Glutamate sensor	[62]
CutRS	*S. lividans*	Actinorhodin biosynthesis repression	[63]

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
