# Peer review of "Regulation of Protein Post-Translational Modifications on Metabolism of Actinomycetes"

_biomolecules, 2020, doi:10.3390/biom10081122_

Round 1

Reviewer 1 Report

The review of Sun et al., summarizes protein post-transaltional modifications (PTM) in actinomycetes with a special emphasis on phosphorylation, acylation and pupylation.

In my opinion the review article targets an attractive topic, which would be of interest for the scientific community working with actinobacterial natural compound producers. However, the way the manuscript is currently written and composed it is not suitable to be published. The manuscript suffers from an inappropriate English and scientific writing style. A lot of text passages include general statements without real scientific content, which makes it difficult for the reader to recognize the scientific context (see below). Several important references are missing and the reference list is full of mistakes, which may serve as an indicator of careless work.

Unfortunately, there are no line numbers in the manuscript, so I start with line number 1 at the „Abstract“ and then again with „1“ on the top of each page.

Abstract

Line 3-4: weird accessory sentence “,especially in the soil,”; should be “environmental conditions”

Line 7: the wording “signal transportation” is misleading here

Line 12: rewrite “advance of PTMs”

Introduction

The Introduction part misses relevant information. It is more a general text but does not contribute for a better understanding of the role and function of PTMs in actinomycetes.

Page 1

Line 16: should be “secondary metabolism”

Line 16-17: sentence is too imprecise. Define “It”. At the moment the authors talk about actinomycetes in general, now about an individual species.

Line 17-18: rewrite “Not it…” see comment above

Line 21: rewrite “treasure for bioactive natural products”. What is meant here? Treasure chest? Or “important source for,…”

Line 22-23: What is meant with “Most natural products are secondary metabolites”? Secondary metabolites are natural products…

Line 24-28: please correct in terms of proper English writing. “contain”. Furthermore, these sentences should be optimized for more preciseness. It reads like an unkind compilation of, however, relevant background information. Furthermore, relevant review articles should be cited here.

Page 2

Line 1-3: Too imprecise: Which nutrients are released? What means at the expense of hyhpa?

Line 6-8: The authors write three processes but mention 4. Should be clarified

Line 11-12: “to highly expand” should be “which highly expands…”

Line 14-16: This sentence needs citations for the various mentioned cellular processes and also some precise examples.

Line 19-20: If the function of PTMs in primary metabolism are known, then it should be cited here and some examples should be given, otherwise the reader does not have any benefit from reading this sentence.

Line 21-23: Uninformative sentence. What has been shown in this article?

Line 23-24: should be “for S. coelicolor” Uncomplete sentence. Something is missing here.

Line 26-26: incorrect writing “natural products in Actinomycetes are achieved…”

Line 29: inadequate sentence “Phosphoproteome is the best illustrated PTMs…”

From here on I stop correcting or highlighting individual sentences but comment only on the general content of the manuscript. However, I would like to point out that the manuscript needs optimization in terms of English and scientific writing style.

Line 42-43: References are missing for the individual effects on gene function.

Line 46-47: The different kind of kinases should be mentioned and explained how they work.

Page 3,

line 9-12: Reference missing.

Line 45: How do the response regulators including diguanylate cyclase and methyltransferase work?

Line 36: To what does the conservation refers to? Genetic or protein level?

Line 37-38: What’s the clear activation signal and the target genes? This is way to imprecise. The authors need to present the molecular principles more in detail otherwise the reader has no change to learn anything valuable from this text. It is also no appropriate to only refer to citations and force the reader to find these data by himself but instead everything need to be outlined clearly in the text and should be supported by citations, figures and tables. The information on the target genes is given then sometime later on in line 49-50. Here the according reference is missing. This is one example where the manuscript should be optimized in terms of a better organization.

Page 4

Line 21-24: references missing

Line 25-30: How does the PknB structure look like? What’s the function of the activation loop? What is the PknB ligand?

Line 29-31: What does the information of the heterodimers help? What’s the conclusion here?!

Line 44-45: reference missing

Page 5 - Acetylation/deacetylation paragraph: Only the first paragraph refers to processes in actinomycetes, whereas the rest reports on general mechanisms in prokaryotes, B. subtilis and E. coli. Thus, there is not much benefit for a better understanding about this process in actinomycetes. I understand that the authors first would like to introduce the (de)acetylation process. However, this is also not done for the other PTMs. Should be consistent…

Page 6

Line 25-27: reference missing

Line 46-48: organizaton: S. erythraea was mentioned before and thus should have been introduced as erythromycin producer before.

Line 50-next page: reference missing

Page 7

Line 6: the gene was knocked out – not the protein! Incorrect denomination of term

Line 24: Which researchers?

Page 8

Line19-20: Unclear sentence on acyl-CoAs. Acyl-CoAs are also natural compounds

Line 27: How does the metabolic engineering based on PTMs look like? Should be exlplained!

Line 29-30 rewrite “regulatory mechanisms of where acyl-CoAs flow”

Line 30-31: Non-significant sentence

References

The reference list is not acceptable. It is incomplete in terms of reference information and full of mistakes: e.g. in terms of

- manuscript information - journal name, volume page number is missing for reference 3., 21.-27., 36.-37. and following citations.

- Gene and strains names have to be written in italics (e.g. 3.-4., 7.-9., 21.-23) but also many more)

Incorrect usage of large and small letters, e.g. should be ac(et)yl-CoA-derived (12.; 57.), Ser/Thr/Tyr (13.), GlnR (56.-58.) (40.,41.) and several others

Strange information in (13.)

Incorrect author names, such as Fau in (36., 39., 40.,47.,…)

Again – these are only a few examples, there are many more, which should be corrected.

Figures and Tables

Table 1: Where does the mass change come from?!

Table 2: AfsQ1/Q2 should have citation (Wang et al., 2013); What about GlnR?

Fig.2: should be “Tacrolimus”. What means “Ac-CoA”? should be written in a different manner and/or mentioned in the legend.

Legend misses information, such as explanation for abbreviations AcP, Pta, AcP,… (also true for legend 1)

Furthermore, it is not appropriate to include comments and explanations like “On one hand,…” into the legend. This should be part of the main text. Legends should harbor all the information, which are needed to understand the figure details.

Author Response

Dear Reviewer,

     We all authors appreciate so much to you. The suggestions and comments are extremely helpful for our improvement of manuscript. The point-by-point response to the your comments are uploaded in the word version. Please see the attachment.

    Thank you very much for your attention and consideration.

    Sincerely,

    Xu-Ming Mao

Reviewer 2 Report

This is a detailed study on PTMs involved in Actinomycetes species action.

What is missing ; concluding sentence or two on each mechanism involved.

Also such a work should be translational not only descriptive. So for example which of the PTM's is relevant for using the bacteria to develop effective antibiotics.

 On the other hand when PTM actually confers antibiotic resistance mentioned on EColi and Mtb. 

A specific table should be introduced to delineate specific those bacteria having antibiotic properties and in them which PTM is involved. (not current table which is mixed.

Overall, the conclusion should delineate the approach to be used by those that are working on the field for better understanding and reducing antibiotic resistance. On the other hand which bacteria species based on their properties should be an improved or future target for novel antibiotics development. Currently only Erythromycin is mentioned 

Author Response

(The authors gave the same response as above.)

Reviewer 3 Report

Specific comments are included in the PDF version of the manuscript edited by this reviewer. In general, the paper is not well-written and is poorly organized. It would have been more informative had the authors chosen specific pathways and/or systems in Actinobacteria that are regulated by posttranslational modifications and elaborated the details of those systems. Moreover, the authors did not discuss all of the PTMs listed in their Table 1.

Should the authors choose to resubmit a revised version of this paper, it is strongly suggested that they have their revision edited by a native writer of English before that resubmission.

Author Response

Dear Reviewer,

     We all authors appreciate so much to you. The suggestions and comments are extremely helpful for our improvement of manuscript. The responses to all the comments are answered in the PDF version you attached in comments. Please see the attachment.

   The reason why we did not discuss all of the PTMs listed in their Table 1 is that there are inadequate results in some PTMs identified in Actinobacteria, like O-glycosylation. In Table 1, we summarized all proteome analysis of PTMs in Actinobacteria. However, some of them just revealed which proteins were modified without detailed mechanisms. We hope that we can provide as much comprehensive information as possible to facilitate the follow-up related researches.

    Thank you very much for your attention and consideration.

    Sincerely,

    Xu-Ming Mao

Round 2

Reviewer 1 Report

The major benefit a reader should have from reading a review is a nice organization of data and a good written text, both of which still should be optimized in the current manuscript. Here are only a few minor points, which I noticed, however, the full paper should be revised with a focus on each individual sentence.

Line 15-16: “in the identification”

Line 17: physiological metabolism

Line 22: “developments of PTMs” not the development of PTMs is meant here but the progress of research on PTMs

Line 30: then morphology should be described here

Line 31: rewrite “Now”

Line 41-42: rewrite “better exploitation of natural products and control of pathogenicity”

Line 45: define “them”

Line 62-63: Unclear sentence: “…functions of PTMs … are well-established, such as cell division…”

Line 66: rewrite “concept of PTMs”

Line 73: Unclear writing: “The phosphoproteome is the best illustrated protein modification of the bacterial PTM,…”

Line 82: Inappropriate: “is the earliest type of PTM” What is meant here?

Line 87/95/334: I mentioned this before that it is not good to use the wording “can”

Line 93: Define “its”

Line 101: Inappropriate “TFs use… to achieve its function”

Line 138: Rewrite “It is possible that…”

Line 140-141: Mixing up of terms “gene – protein”

Line 174: should be optimized for more preciseness

Line 178-180: The sentence describes PknB and cites Figure 1A, however PknB is not shown in Figure 1A. Figure 1A shows a schematic presentation of a TCS.

Line 182-184: Rewrite “catalytic site mutants”. What means “consistent with the allosteric activation…” Explain in more detail.

Line 189: correct “PknB mutant kinase”

Line 200: “Rewrite “Autophosphorylation activates its kinase activity”

Line 231: Rewrite “it has been known” Is it known or not?! Or do you mean “It has been shown”?!

Line 235: format; font size

Line 254 and following: Why is this separation done for Acetylation but not for the other PTM mechanisms?! Then why starting with a methodological aspect here?! It’s rather unpleasant to start the sentence with “After enrichment…”

Line 277: activator/repressor of which gene in these organisms?

Line 278: GlnR acts as a transcriptional regulator. Therefore target genes and not the expression products should be mentioned.

Line 296: blank too much in front of “production”

Line 300-301: Inverse logic

Line 309: rewrite “parent nucleus”

Line 342: S. coelicolor

Line 346-347: incorrect writing of mutant

Reference list: Still strain and gene names are not written in italics!

Table 1: blanks missing between “S.” and species name; rewrite “feeder metabolic pathways”

Table 2: rewrite “uptaken”

Figure 1: should be “Tyrosin” in the figure

Author Response

(The authors gave the same response as above.)

Reviewer 3 Report

Comments are provided in the accompanying PDF.

Author Response

Dear Reviewer,

We all authors appreciate so much to you. The suggestions and comments are extremely helpful for our improvement of manuscript. The responses to all the comments are answered in the PDF version you attached in comments. Please see the attachment.

Thank you very much for your attention and consideration.

Sincerely,

Xu-Ming Mao
